# MODEL-BASED REINFORCEMENT LEARNING FOR BIOLOGICAL SEQUENCE DESIGN

**Christof Angermueller**
Google Research
{christofa}@google.com

**David Dohan**
Google Research
{ddohan}@google.com

**David Belanger**
Google Research
{dbelanger}@google.com

**Ramya Deshpande**[*]
Caltech
{rdeshpan}@caltech.edu

**Kevin Murphy**
Google Research
{kpmurphy}@google.com

**Lucy Colwell**
Google Research
University of Cambridge
{lcolwell}@google.com

## ABSTRACT

The ability to design biological structures such as DNA or proteins would have considerable medical and industrial impact. Doing so presents a challenging black-box optimization problem characterized by the large-batch, low round setting due to the need for labor-intensive wet lab evaluations. In response, we propose using reinforcement learning (RL) based on proximal-policy optimization (PPO) for biological sequence design. RL provides a flexible framework for optimization generative sequence models to achieve specific criteria, such as diversity among the high-quality sequences discovered. We propose a model-based variant of PPO, DyNA PPO, to improve sample efficiency, where the policy for a new round is trained offline using a simulator fit on functional measurements from prior rounds. To accommodate the growing number of observations across rounds, the simulator model is automatically selected at each round from a pool of diverse models of varying capacity. On the tasks of designing DNA transcription factor binding sites, designing antimicrobial proteins, and optimizing the energy of Ising models based on protein structure, we find that DyNA PPO performs significantly better than existing methods in settings in which modeling is feasible, while still not performing worse in situations in which a reliable model cannot be learned.

## 1 INTRODUCTION

Driven by real-world obstacles in health and disease requiring new drugs, treatments, and assays, the goal of biological sequence design is to identify new discrete sequences $x$ which optimize some oracle, typically an experimentally-measured functional property $f(x)$. This is a difficult black-box optimization problem over a combinatorially large search space in which function evaluation relies on slow and expensive wet-lab experiments. The setting induces unusual constraints in black-box optimization and reinforcement learning: large synchronous batches with few rounds total.

The current gold standard for biomolecular design is *directed evolution*, which was recently recognized with a Nobel prize (Arnold, 1998) and is a form of randomized local search. Despite its impact, directed evolution is sample inefficient and relies on greedy hillclimbing to the optimal sequences. Recent work has demonstrated that machine-learning-guided optimization (Section 3) can find better sequences faster.

---

[*]Work done as an intern at Google.

Reinforcement learning (RL) provides a flexible framework for black-box optimization that can harness modern deep generative sequence models. This paper proposes a simple method for improving the sample efficiency of policy gradient methods such as PPO (Schulman et al., 2017) for black-box optimization by using surrogate models that are trained online to approximate $f(x)$. Our method updates the policy's parameters using sequences $x$ generated by the current policy $\pi_\theta(x)$, but evaluated using a learned surrogate $f'(x)$, instead of the true, but unknown, oracle reward function $f(x)$. We learn the parameters of the reward model, $w$, simultaneously with the parameters of the policy. This is similar to other model-based RL methods, but simpler, since in the context of sequence optimization, the state-transition model is deterministic and known. Initially the learned reward model, $f'(x)$, is unreliable, so we rely entirely on $f(x)$ to assess sequences and update the policy. This allows a graceful fallback to PPO when the model is not effective. Over time, the reward model becomes more reliable and can be used as a cheap surrogate, similar to Bayesian optimization methods (Shahriari et al., 2015). We show empirically that cross-validation is an effective heuristic for assessing the model quality, which is simpler than the inference required by Bayesian optimization.

We rigorously evaluate our method on three *in-silico* sequence design tasks that draw on experimental data to construct functions $f(x)$ characteristic of real-world design problems: optimizing binding affinity of DNA sequences of length 8 (search space size $4^8$); optimizing anti-microbial peptide sequences (search space size $20^{50}$), and optimizing binary sequences where $f(x)$ is defined by the energy of an Ising model for protein structure (search space size $20^{50}$). These do not rely on wet lab experiments, and thus allow for large-scale benchmarking across a range of methods. We show that our DyNA PPO method achieves higher cumulative reward for a given budget (measured in terms of number of calls to $f(x)$) than existing methods, such as standard PPO, various forms of the cross-entropy method, Bayesian optimization, and evolutionary search.

In summary, our contributions are as follows:

- We provide a model-based RL algorthm, DyNA PPO, and demonstrate its effectiveness in performing sample efficient batched black-box function optimization.

- We address model bias by quantifying the reliability and automatically selecting models of appropriate complexity via cross-validation.

- We propose a visitation-based exploration bonus and show that it is more effective than entropy-regularization in identifying multiple local optima.

- We present a new optimization task for benchmarking methods for biological sequence design based on protein energy Ising models.

## 2 METHODS

Let $f(x)$ be the function that we want to optimize and $x \in V^T$ a sequence of length $T$ over a vocabulary $V$ such as DNA nucleotides ($|V| = 4$) or amino acids ($|V| = 20$). We assume $N$ experimental rounds and that $B$ sequences can be measured per round. Let $D_n = \{(x, f(x))\}$ be the data acquired in round $n$ with $|D_n| = B$. For simplicity, we assume that the sequence length $T$ is constant, but our approach based on generating sequences autoregressively easily generalizes to variable-length sequences.

### 2.1 MARKOV DECISION PROCESS

We formulate the design of a single sequence $x$ as a Markov decision process $\mathcal{M} = (S, A, p, r)$ with state space $S$, action space $A$, transition function $p$, and reward function $r$. The state space $S = \cup_{t=1...T} V^t$ is the set of all possible sequence prefixes and $A$ corresponds to the vocabulary $V$. A sequence is generated left to right. At time step $t$, the state $s_t = a_0, ..., a_{t-1}$ corresponds to the $t$ last tokens and the action $a_t \in A$ to the next token. The transition function $p(s_t + 1|s_t) = s_t a_t$ is deterministic and corresponds to appending $a_t$ to $s_t$. The reward $r(s_t, a_t)$ is zero except at the last step $T$, where it corresponds to the functional measurement $f(s_{T-1})$. For generating variable-length sequences, we extend the vocabulary by a special end-of-sequence token and terminate sequence generation when this token is selected.

---

Algorithm 1: DyNA PPO

---

1: **Input:** Number of experiment rounds **N**
2: **Input:** Number of model-based training rounds **M**
3: **Input:** Set of candidate models $\mathcal{S} = \{f'\}$
4: **Input:** Minimum model score $\tau$ for model-based training
5: **Input:** Policy $\pi_\theta$ with initial parameters $\theta$
6: **for** $n = 1, 2, ...\mathcal{N}$ **do**
7:     Collect samples $\mathcal{D}_n = \{x, f(x)\}$ using policy $\pi_\theta$
8:     Train policy $\pi_\theta$ on $\mathcal{D}_n$
9:     Fit candidate models $f' \in \mathcal{S}$ on $\bigcup_{i=1}^n \mathcal{D}_i$ and compute their score by cross-validation
10:     Select the subset of models $S' \subseteq S$ with a score $\geq \tau$
11:     **if** $\mathcal{S}' \neq \emptyset$ **then**
12:         **for** $m = 1, 2, ...\mathbf{M}$ **do**
13:             Sample a batch of sequences $x$ from $\pi_\theta$ and observe the reward $f''(x) = \frac{1}{|\mathcal{S}'|} \sum_{f' \in \mathcal{S}'} f'(x)$
14:             Update $\pi_\theta$ on $\{x, f''(x)\}$
15:         **end for**
16:     **end if**
17: **end for**

---

## 2.2 POLICY OPTIMIZATION

We train a policy $\pi_\theta(a_t|s_t)$ to optimize the expected sum of rewards :

$$\mathbb{E}[R(s_{1:t})|s_0, \theta] = \sum_{s_t} \sum_{a_t} \pi_\theta(a_t|s_t)r(s_t, a_t). \tag{1}$$

We use *proximal policy optimization* (PPO) with KL trust-region constraint (Schulman et al., 2017), which we have found to be more stable and sample efficient than REINFORCE (Williams, 1992). We have also considered off-policy deep Q-learning (DQN) (Mnih et al., 2015), and categorical distributional deep Q-learning (CatDQN) (Bellemare et al., 2017), which are in principle more sample-efficient than on-policy learning using PPO since they can reuse samples multiple times. However, they performed worse than PPO in our experiments (Appendix C). We implement algorithms using the TF-Agents RL library (Guadarrama et al., 2018).

We employ autoregressive models with one fully-connected layer as policy and value networks since they are faster to train and outperformed recurrent networks in our experiments. At time step $t$, the network takes as input the $W$ last characters $a_{t-W}, ..., a_{t-1}$ that are one-hot encoded, where the context window size $W$ is a hyper-parameter. To provide the network with information about the current position of the context window, it also receives the time step $t$, which is embedded using a sinusoidal positional encoding (Vaswani et al., 2017), and concatenated with the one-hot characters. The policy network outputs a distribution $\pi_\theta(a_t|s_t)$ over next the token $a_t$. The value network $V(s_t)$, which approximates the expected future reward for being in state $s_t$, is used as a baseline to reduce the variance of stochastic estimates of equation 1 (Schulman et al., 2017).

## 2.3 MODEL-BASED POLICY OPTIMIZATION

Model-based RL learns a model of the environment that is used as a simulator to provide additional pseudo-observations. While model-free RL has been successful in domains where interaction with the environment is cheap, such as those where the environment is defined by a software program, its high sample complexity may be unrealistic for biological sequence design. In model-based RL, the MDP $\mathcal{M} = (S, A, p, r)$ is approximated by a model $\mathcal{M}' = (S, A, p', r')$ with the same state space $S$ and action space $A$ as $\mathcal{M}$ (Sutton & Barto, 2018, Ch. 8). Since the transition function $p$ is deterministic in our case, only the reward function $r(s_t, a_t)$ needs to be approximated by $r'(s_t, a_t)$. Since $r(s_T, a_T)$ is non-zero at the last step $T$ and then corresponds to $f(x)$ with $x == s_{T-1}$, the problem reduces to approximating $f(x)$. This can be done by supervised regression by fitting a regressor $f'(x)$ on the data $\cup_{n' <= n} D_{n'}$ collected so far. We then use the resulting model to collect additional observations $(x, f'(x))$ and update the policy in a simulation phase, instead of only using observations $(x, f(x))$ from the the true environment, which are expensive to collect. We call our method DyNA PPO since it is similar to the DYNA architecture (Sutton (1991); Peng et al. (2018)) and since can be used for DNA sequence design.

Model-based RL provides the promise of improved sample efficiency when the model is accurate, but it can reduce performance if insufficient data are available for training a trustworthy model. In this case, the policy is prone to exploit regions where the model is inaccurate (Janner et al., 2019). To reap the benefit of model-based RL when the model is accurate and avoid reduced performance when it is not, we (i) automatically select the model from a set of candidate models of varying complexity, (ii) only use the selected model if it is accurate, and iii) stop model-based training as soon the the model uncertainty increases by a certain threshold. After each round of experiment, we fit a set of candidate models on all available data to estimate $f(x)$ via supervised regression. We quantify the accuracy of each candidate model by the $R^2$ score, which we estimate by five-fold cross-validation. See Appendix G for a discussion of different data splitting strategies to select models using cross-validation. If the $R^2$ score of all candidate model is below a pre-specified threshold $\tau$, we do not perform model-based training in that round. Otherwise, we build an ensemble model that includes all models with a score greater or equal than $\tau$, and use the average prediction as reward for training the policy. We considered $\tau$ as a tunable hyper-parameter, were we found $\tau = 0.5$ to be optimal for all problems (see Figure 14. By ignoring the model if it is inaccurate, we aim to prevent the policy from exploiting deficiencies of the model (Janner et al., 2019).

We perform up to $M$ model-based optimization rounds (see Algorithm 1) and stop as soon as the model uncertainty increased by a certain factor relative to the model uncertainty at the first round ($m = 1$). This is motivated by our observation that the model uncertainty is strongly correlated with the unknown model error, and prevents from training the policy with inaccurate model predictions (see Figure 12, 13) as soon as the model starts to explore regions on which the model was not trained on.

For models, we consider nearest neighbor regression, Bayesian ridge regression, random forests, gradient boosting trees, Gaussian processes, and ensemble of deep neural networks. Within each model family, we additionally use cross-validation for tuning hyper-parameters, such as the number of trees, tree depth, kernels and kernel parameters, or the number of hidden layers and units (see Appendix A.7 for details). By testing and optimizing the hyper-parameters of different models automatically, the model capacity can dynamically increase as data becomes available.

In Bayesian optimization, non-parametric models such as Gaussian processes are popular regressors, and they also automatically grow model capacity as more data arrives (Shahriari et al., 2015). However, with Bayesian optimization there is no opportunity to ignore the regressor entirely if it is unreliable. Furthermore, Bayesian optimization relies on performing (approximate) Bayesian inference, which in practice is sensitive to the choice of hyper-parameter (Snoek et al., 2012).

Overall, our method combines the positive attributes of both generative and discriminative approaches to sequence design. Our experiments do not compare to prior work on model-based RL, since these methods primarily focus on estimating a dynamics model for state transitions.

## 2.4 Diversity-Promoting Reward Function

Learning policies to generate diverse sequences is important because of several reasons. In many applications, $f(x)$ is an *in-vitro* (taking place outside a living organism) surrogate for an *in-vivo* taking place inside a living organism) functional measurement that is even more expensive to evaluate than $f(x)$. The in-vivo measurement may depend on properties that are correlated with $f(x)$ and others that are not captured at all in-vitro, such as off-target effects or toxicity. To improve the chance that a sequence satisfying the ultimate in-vivo criteria is found, it is therefore desirable for the optimization procedure to discover a diverse set of candidate optima. Here, diversity is a downstream metric, for which training the policy $\pi_\theta(x)$ to maximize equation 1 will not necessarily yield good performance. For example, a high-quality policy can learn to always generate the same sequence $x$ with a high value of $f(x)$, and will therefore result in zero diversity. An additional reason that diversity matters is that it yields a good exploration strategy, even for scenarios where optimizing equation 1 is sufficient. Finally, use of strategies that reward high-diversity policies can reduce the policies' tendency to generate exact duplicates.

To increase sequence diversity, we employ a simple exploration reward bonus based on the density of proposed sequences, similar to existing exploration techniques based on state visitation frequency (Bellemare et al., 2016). Specifically, we define the final reward as $r_T = f(x) - \lambda \cdot \text{dens}_\epsilon(x)$, where $\text{dens}_\epsilon(x) \in \mathbb{N}^+$ is the weighted number of sequences that have been proposed in previous

rounds with a distance of less than $\epsilon$ away from $x$, where the weight decays linearly with the distance. This reward penalizes proposing similar sequences multiple times, where the strength of the penalty is controlled by $\lambda$. As a result, the policy learns not to generate related sequences and hence explores the search space more effectively. We used the edit distance as distance metric and tuned the distance radius $\epsilon$, where setting $\epsilon > 0$ improved exploration on high-dimensional problems (see Figure 11). We also considered an alternative penalty based on the nearest neighbor distance of the proposed sequence to past sequences, which we found to be less effective (see Figure 9).

## 3 RELATED WORK

Recently, machine learning approaches have been shown to be effective in optimizing real-world DNA and protein sequences (Wang et al., 2019; Chhibbar & Joshi, 2019; de Jongh et al., 2019; Liu et al., 2019; Sample et al., 2019; Wu et al., 2019). Existing methods for biological sequence design fall into three broad categories: evolutionary search, optimization using discriminative models (e.g. Bayesian optimization), and optimization using generative models (e.g. the cross entropy method).

Evolutionary approaches perform direct local search in the space of sequences. They include the aforementioned directed evolution and derivatives with application-specific mutation and recombination steps. Evolutionary approaches are appealing since they are simple and can easily incorporate human intuition into the design process, but generally suffer from low sample efficiency.

Optimization methods based on discriminative models alternate between two steps: (i) using the data that have been collected so far to fit a regressor $f'(x)$ to approximate $f(x)$, and (ii) using $f'(x)$ to define an *acquisition function* that is optimized to select the next batch of sequences. Recently, such an approach was used to optimize the binding affinity of IgG antibodies (Liu et al., 2019), where a neural network ensemble was used for $f'(x)$. In general, optimizing the acquisition function is a non-trivial combinatorial optimization problem. Liu et al. (2019) employed *activation maximization*, where gradient-based optimization is performed on a continuous relaxation of the discrete search space. However, this requires $f'(x)$ to be differentiable and optimization of a continuous relaxation is vulnerable to leaving the data manifold (cf. deep dream (Mordvintsev et al., 2015)).

*Bayesian optimization* defines an acquisition function such as the *expected improvement* (Mockus et al., 2014) based on the uncertainty of $f'(x)$, which enables balancing exploration and exploitation (overview provided in Shahriari et al. (2015)). Gaussian processes (GPs) are commonly used for Bayesian black-box optimization since they provide calibrated uncertainty estimates. Unfortunately, GPs are hard to scale to large, high-dimensional datasets and are sensitive to the choice of hyperparameters. In response, recent work has performed continuous black-box optimization in the latent space of a deep generative model (Gómez-Bombarelli et al., 2018). However, this approach requires a pre-trained model such as a variational autoencoder to obtain the latent embeddings. Our model-based reinforcement learning approach is similar to these approaches in that we train a reinforcement learning policy to optimize a model $f'(x)$. However, our policy is also trained directly on observations of $f(x)$ and is able to resort to model-free training by automatically identifying if the model $f'(x)$ is too inaccurate to be used as surrogate of $f(x)$. Janner et al. (2019) investigated conditions in which an estimate of model generalization (their analysis uses validation accuracy) could justify model usage in such model-based policy optimization settings. Hashimoto et al. (2018) proposed using a cascade of classifiers, one per round, to guide sampling progressively better candidates.

Optimization methods based on generative models seek to learn a distribution $p_\theta(x)$ parameterized by $\theta$ that maximizes the expected value of $f(x)$: $\mathbb{E}_{x \sim P_\theta(x)}[f(x)]$. We note that this is the same form as variational optimization objectives, which allow the use of parameter-space evolutionary strategies (Staines & Barber, 2013; Wierstra et al., 2014; Salimans et al., 2017). Variants of the *cross entropy method* (De Boer et al., 2005; Brookes et al., 2019a) optimize $\theta$, by alternating two steps: (i) sampling $x \sim p_\theta(x)$ and evaluating f(x), and (ii) updating $\theta$ to maximize this expectation. Methods differ in how step (ii) is performed. For example, hillclimb-MLE (Neil et al., 2018) performs maximum-likelihood training on the top k sequences from step (i). Similarly, Feedback GAN (FBGAN) uses samples whose target function value $f(x)$ exceeds a fixed threshold for training a generative adversarial network (Gupta & Zou, 2018). Design by Adaptive Sampling (DbAs) performs weighted MLE of variational autoencoders (Kingma & Welling, 2014), where a sample's weight corresponds to the probability that $f(x)$ is greater than a quantile cutoff under an noise

model (Brookes & Listgarten, 2018). In Brookes et al. (2019b), $p_\theta(x)$ is further restricted to stay close to a prior distribution over sequences.

An alternative approach for optimizing the above expectation is RL. While RL has been used for generating natural text (Bahdanau et al., 2016), small molecules (Zhou et al., 2019), and RNA sequences that fold into a particular structure (Runge et al., 2018), we are not aware of applications of RL to optimizing DNA and protein sequences.

DyNA PPO is related to existing work on model-based RL for sample efficient control (Deisenroth & Rasmussen, 2011; Kurutach et al., 2018; Peng et al., 2018; Kaiser et al., 2019; Janner et al., 2019), with the key difference that the state transition function is known and the reward function is unknown in our work, whereas most existing model-based RL approaches seek to model the state-transition function and consider the reward function as known.

Prior work on sequence generation incorporates non-differentiable rewards, like BLEU in machine translation, via weighted maximum likelihood (MLE). Norouzi et al. (2016) introduce reward augmented MLE, while Bahdanau et al. (2016) fine tune an MLE-pretrained model using actor-critic methods. Reinforcement learning has also been applied to solving combinatorial optimization problems (Bello et al., 2016; Bengio et al., 2018; Dai et al., 2017; Kool et al., 2018). In this setting sample complexity is less important because evaluating $f(x)$ only involves a fast software program.

Recent work has proposed generative models of protein structures (Sabban & Markovsky, 2019) or generative models of amino acids conditional on protein structure (Ingraham et al., 2019). Such methods are outside of the scope of this paper's experiments, since they could only be used in experimental settings where protein structures, which are expensive to measure, are available.

Finally, DNA and protein design differs from small molecule design (Griffiths & Hernández-Lobato, 2017; Kusner et al., 2017; Gómez-Bombarelli et al., 2018; Jin et al., 2018; Sanchez-Lengeling & Aspuru-Guzik, 2018; Korovina et al., 2019) in the following points: (i) the number of sequences measured in parallel in the lab is typically higher (hundred or thousands vs. dozens) due to the maturity of DNA synthesis and sequencing technology, (ii) the search space is a set of sequences instead of molecular graphs, which require specialized network architectures for both discriminative and generative models, and (iii) molecules must be optimized subject to the constraint that there is a set of reactions to synthesize them, whereas practically all DNA or protein sequences are synthesizable.

## 4 EXPERIMENTS

In the next three sections, we compare DyNA PPO to existing methods on three *in-silico* optimization problems that we designed in collaboration with life scientists to faithfully simulate the behavior of real wet-lab experiments, which would be cost prohibitive for a comprehensive methodological evaluation. Along the way, we present ablation experiments to help to better understand the behavior of DyNA PPO.

We compare the performance of model-free policy optimization (**PPO**) and model-based optimization (**DyNA PPO**) with the following methods that we discussed in Section 3. Further details for each method can be found in Appendix A:

- **RegEvolution**: Local search based on regularized evolution (Real et al., 2019), which has performed well on other black-box optimization tasks and can be seen as an instance of directed evolution.

- **DbAs**: Cross-entropy optimization using variational autoencoders (Brookes & Listgarten, 2018).

- **FBGAN**: Cross entropy optimization using generative adversarial networks (Gupta & Zou, 2018).

- **Bayesopt GP**: Bayesian optimization using a Gaussian process regressor and activation maximization as acquisition function solver.

- **Bayesopt ENN** Bayesian optimization using an ensemble of neural network regressors and activation maximization as acquisition function solver.

      • **Random**: Guessing sequences uniformly at random.

We quantify optimization performance by the cumulative maximum reward $f(x)$ for sequences proposed up to a given round, and we use the area under the cumulative maximum reward curve to summarize one optimization trajectory as a single number. We quantify sequence diversity (Section 2.4) in terms of the mean pairwise hamming distance between the sequences proposed at each round. For problems with known optima, we also report the fraction of global optima found. We replicate experiments with 50 random seeds.

## 4.1 Optimization of Protein Contact Ising Models

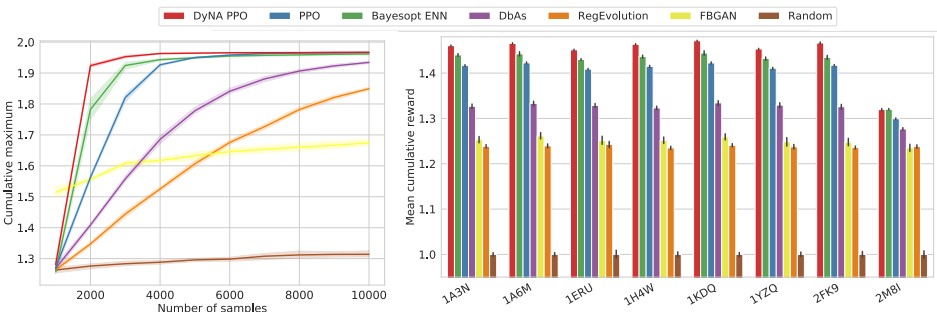

Figure 1: **Comparison of methods on optimizing the energy of protein contact Ising models**. Left: the cumulative maximum reward depending on the number of rounds for one selected protein target (1A3N). Right: the mean cumulative maximum relative to *Random* for alternative protein targets. Since $f(x)$ can be well-approximated by a model trained on few examples, model-based training (DyNA PPO) results in a clear improvement over model-free training (PPO).

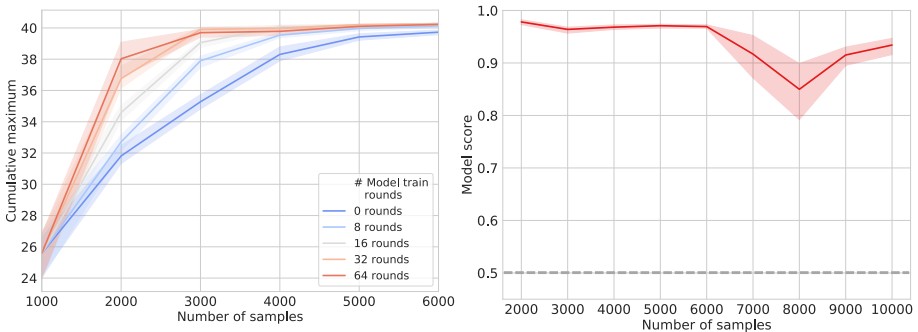

Figure 2: **Analysis of the performance of DyNA PPO on the Ising model.** Left: Performance of DyNA PPO depending on the number of inner policy optimization rounds using the surrogate model. Using 0 rounds corresponds to PPO training. Since the surrogate model is sufficiently accurate, it is useful to perform many inner loop optimization rounds before querying $f(x)$ again. Right: the $R^2$ of the surrogate model. Since it is always above the threshold for model-based training (0.5; dashed line), it is always used for training.

We first consider synthetic black-box optimization problems based on the 3D structure of naturally-occurring proteins. Ising models fit on sets of evolutionary-related protein sequences have been shown to be accurate predictors for proteins' 3D structure (Shakhnovich & Gutin, 1993; Weigt et al., 2009; Marks et al., 2011; Sułkowska et al., 2012). We consider the inverse problem: given a protein, we seek to find the amino acid sequence that minimizes the energy of the Ising model parameterized by its structure. Optimizers are given a budget of 10 rounds with batch size 1000 and we consider sequences of length 50 (search space size $20^{50}$). The functional form of the energy function is given in Appendix B.1.

On the left of Figure 1 we consider the optimization trajectory for a representative protein and on the right we compare the best $f(x)$ found for each method across a range of proteins. We find that DyNA PPO considerably outperforms the other methods. We expect that this is because this synthetic

reward landscape can be well-described by a model fit using few examples, which also explains the good performance of Bayesian optimization. On the left of Figure 2 we vary the number of inner-loop policy optimization rounds with observations from the model-based environment, where using 0 rounds corresponds to performing standard PPO. Since the surrogate model is of sufficient accuracy already at the beginning (right plot), performing more inner policy optimization rounds increases performance and enables DyNA PPO to generate high-quality sequences using very few evaluations of $f(x)$.

## 4.2 OPTIMIZATION OF TRANSCRIPTION FACTOR BINDING SITES

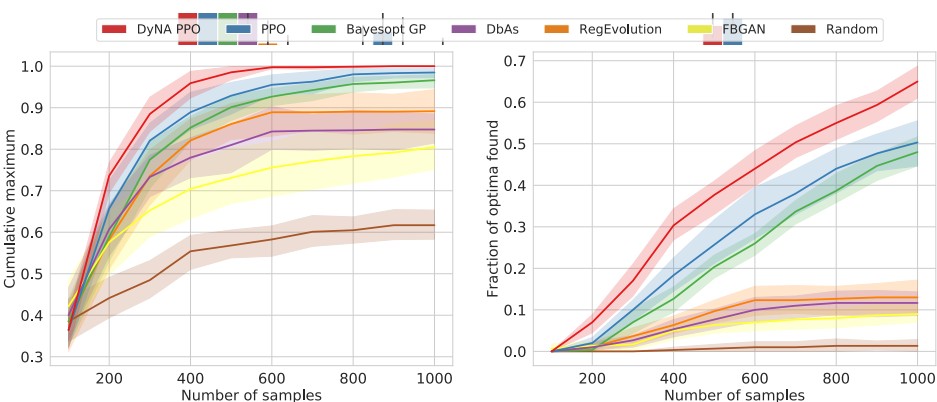

Figure 3: **Comparison of methods on optimization transcription factor binding sites.** Left: maximum cumulative reward $f(x)$ as a function of samples. Right: fraction of local optima found. DyNA PPO optimizes $f(x)$ faster and finds more optima than PPO and baseline methods. Results are shown for one representative transcription factor target (SIX6 REF R1).

|  | DyNA PPO | PPO | BO-GP | DbAs | RegEvol | FBGAN | Random |
|---|---|---|---|---|---|---|---|
| Cumulative maximum | **6.4** | 5.8 | 5.0 | 3.7 | 3.7 | 2.2 | 1.3 |
| Fraction optima found | **6.8** | 5.6 | 5.4 | 3.3 | 3.3 | 2.5 | 1.0 |
| Mean hamming distance | **5.6** | 5.4 | 4.0 | 2.5 | 1.0 | 2.5 | 7.0 |

Table 1: **Mean rank of methods across transcription factor binding targets.** Mean rank of methods across all 41 hold-out transcription factor targets. Ranks were computed within each target using the average of metrics across optimization rounds, and then averaged across target. The higher the rank the better. 7 is the maximum rank. DyNA PPO outperforms the other methods on both optimization of $f(x)$ and its ability to identify multiple well-separated local optima.

Transcription factors are protein sequences that bind to DNA sequences and regulate their activity. Barrera et al. (2016) measured the binding affinity of numerous transcription factors against all possible length-8 DNA sequences ($V = 4$). The resulting dataset defines 158 different discrete optimization tasks, where the goal of each task is to find a DNA sequence of length eight that maximizes the affinity towards one of the transcription factors. It is well suited for in-silico benchmarking since (i) it is exhaustive and thereby does not require estimating missing $f(x)$ and (ii) the distinct local optima of all tasks are known and can be used to quantify exploration (see Appendix B.2 for details). The optimization methods are given a budget of 10 rounds with a batch size of $B = 100$ sequences. The search space size is $4^8$. We use one task (CRX REF R1) for optimizing the hyper-parameters of all methods, and test performance on 41 heterogeneous hold-out tasks.

Figure 3 plots the performance of methods on a single representative binding target (SIX REF R1) as a function of the total number of sequences measured so far. We find that DyNA PPO and PPO outperform all other methods in terms of both the cumulative maximum $f(x)$ found as well as the fraction of local optima discovered. We also find that the diversity of proposed sequences quantified by the fraction of global optima found is high compared to other generative approaches. This shows that our method continues to explore the search space by proposing novel sequences instead of converging to a single sequence or a handful of sequences–a desired property as discussed

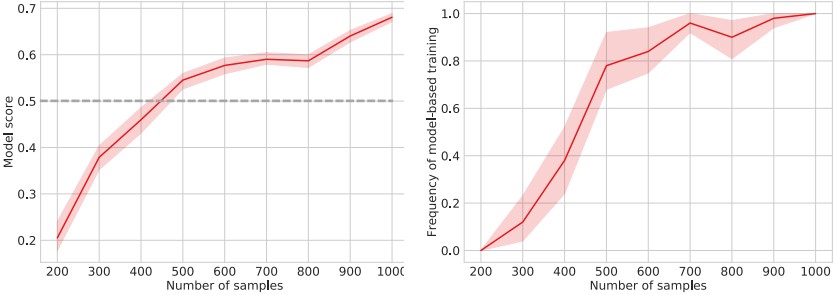

Figure 4: **Analysis of the progression of model-based training on the transcription factor task.** Left: the mean $R^2$ model score averaged across replicas as a function of the number of training samples. The horizontal dashed line indicates the minimum threshold (0.5) for model-based training. Right: the fraction of replicates that performed model-based training based on this threshold. Shows that models tend to be inaccurate in early rounds and are therefore not used for model-based training. This explains the relatively small improvement of DyNA PPO over PPO in Figure 3.

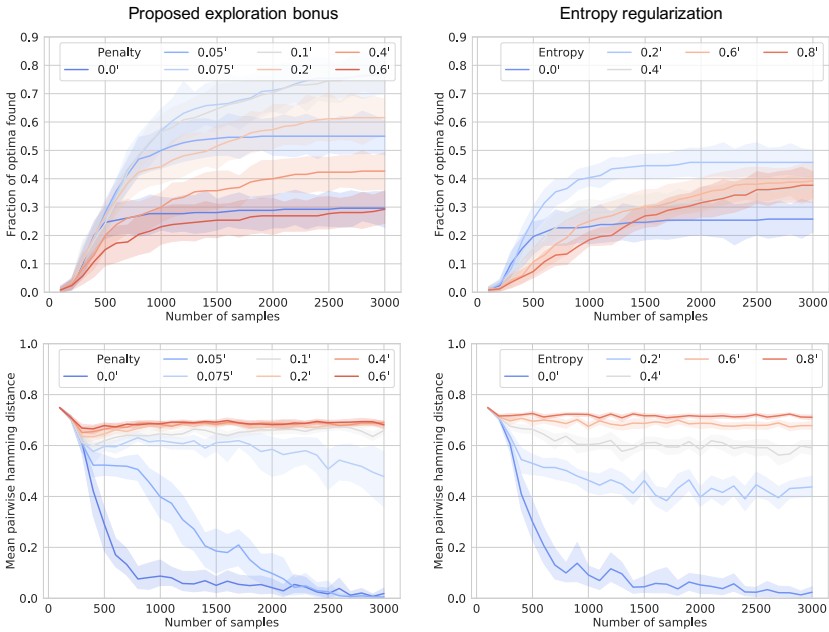

Figure 5: **Comparison of the proposed exploration bonus vs. entropy regularization on the transcription factor task**. Left: performance with exploration bonus as a function of the density penalty $\lambda$ (Section 2.4). Right: performance of entropy regularization as a function of the regularization strength. The top row shows that PPO finds about 80% of local optima with a relatively mild density penalty of $\lambda = 0.1$, whereas only about 45% local optima are found when using entropy regularization. The bottom row shows that varying the density penalty enables to control the sequence diversity quantified by the mean pairwise hamming distance between sequences.

in Section 2.4. Across all tasks DyNA PPO and PPO rank highest compared with other methods (Table 1).

In Figure 4 and 5, we analyze the effects two key design decisions of DyNA PPO: model-based training and promoting exploration. We find that automated model selection automatically increases the complexity of the model, but that the models are not always accurate enough to be used for model-based training. This explains the relatively small improvement of DyNA PPO over PPO. We also find that the exploration bonus outlined in Section 2.4 is more effective than entropy regularization in finding multiple local optima and promoting sequence diversity.

### 4.3 OPTIMIZATION OF ANTI-MICROBIAL PEPTIDES

Next, we seek to design antimicrobial peptides (AMPs). AMPs are relatively short (8 - 75 amino acids) protein sequences ($|V| = 20$ amino acids), which are promising candidates against multi-resistant pathogens due to their wide range of antimicrobial activities. We use the dataset proposed by Witten & Witten (2019), which contains 6,760 unique AMP sequences and their antimicrobial activity towards multiple pathogens. We follow Witten & Witten (2019) for preprocessing the dataset and generating non-AMP sequences as negative training samples. Unlike the transcription factor binding site dataset, we do not have wet-lab experiments for every sequence in the search space. Therefore, we fit random forest classifiers to predict if a sequence is antimicrobial towards a certain pathogen in the dataset (see Section B.3), and use the predicted probability as the functional measurement $f(x)$ to optimize. Given the high accuracy of the classifiers (cross-validated AUC 0.94 and 0.99), we expect that the reward landscape of $f(x)$ is of realistic difficulty. We perform 8 rounds with a batch size 250 and restrict the sequence length to at most 50 characters (search space size $20^{50}$).

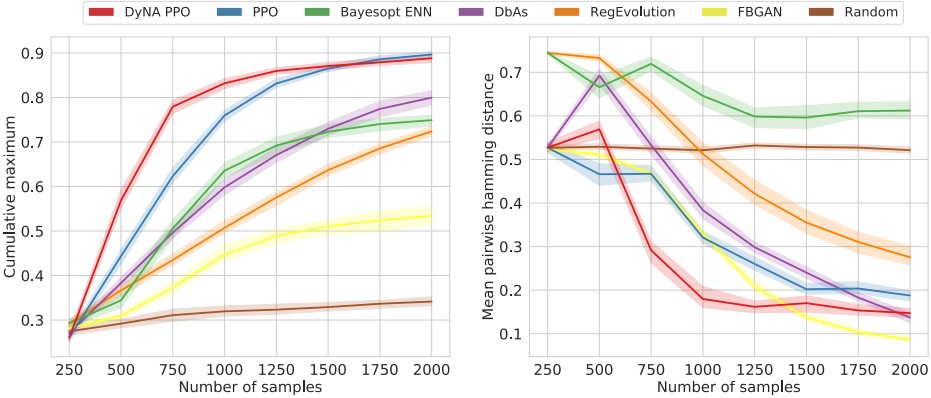

Figure 6: **Comparison of methods on the AMP design task.** Left: Model-based training using DyNA PPO and model-free PPO clearly outperform the other methods in terms of the maximum cumulative reward. Right: The mean pairwise hamming distance between sequences proposed at each round, which is lower for DyNA PPO and PPO but does not converge to zero due to the density-based exploration bonus (Figure 11).

Figure 6 compares methods on C. alibicani. We find that model-based optimization using DyNA PPO enables finding high reward sequences in early rounds, though model-free PPO slightly surpasses the performance of DyNA PPO later on. Both DyNA PPO and PPO considerably outperform the other methods in terms of the maximum $f(x)$ found. The density based exploration bonus prevents PPO and DyNA PPO from generating non-unique sequences (Figure 11). Stopping model-based training as soon as the model uncertainty increased by a certain factor prevents DyNA PPO from converging to a sub-optimal solution when performing many model-based optimization rounds (Figure 12,13).

## 5 CONCLUSION

We have shown that RL is an attractive alternative to existing methods for designing DNA and protein sequences. We have proposed DyNA PPO, a model-based extension of PPO (Schulman et al., 2017) with automatic model selection that improves sample efficiency, and incorporates a reward function that promotes exploration by penalizing identical sequences. By approximating an expensive wet-lab experiment with a surrogate model, we can perform many rounds of optimization in simulation. While this work has been focused on showing the benefit of DyNA PPO for biological sequence design, we believe that the large-batch, low-round optimization setting described here may well be of general interest, and that model-based RL may be applicable in other scientific and economic domain.

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

# A   IMPLEMENTATION DETAILS

## A.1   REGULARIZED EVOLUTION

Regularized evolution is a variant of directed evolution that regularizes the search by keeping a fixed number of individuals alive as candidates for selection (analogous to death by aging). At each round, it generates a batch of child sequences by sampling two parent sequences per child from the population via tournament selection, i.e. selecting the fittest out of K randomly sampled individuals. It then performs crossover of the two parent sequences by copying the characters of one parent from left to right and randomly transitioning to transcribing from the other parent sequence with some crossover probability at each step. Child sequences are mutated by substituting characters independently by other characters with some substitution probability. For variable-length sequences, we also allowed insertion and deletion mutations. As hyper-parameters, we tune the tournament size, substitution-, insertion-, and deletion-probabilities.

## A.2   MCMC AND SIMULATED ANNEALING

MCMC and simulated annealing (Kirkpatrick et al., 1983) resemble evolution with no crossover, and selection only occurring between an individual and its parent. Beginning with a random population, each individual evolves as a single chain, with neighborhood structure defined by the mutation operator described in section A.1. We denote $x$ and $x'$ as a parent and child sequence, respectively. A transition $x \rightarrow x'$ is always accepted if the reward increases ($f(x') > f(x)$). Otherwise, the transition is accepted with some acceptance probability. For MCMC, the acceptance probability is $f(x')/f(x)$, while for simulated annealing it is $\exp((f(x') - f(x))/T)$ for some temperature $T$. A high temperature increases the likelihood of accepting a move that decreases the reward. The next mutation on the chain begins from $x$ if the transition is rejected, and from $x'$ otherwise. We treated the temperature $T$ as a tunable hyper-parameter in addition to the evolution hyper-parameters described in section A.1.

## A.3   FEEDBACK GAN

We follow the methodology suggested by Gupta & Zou (2018). Instead of using a constant threshold for selecting positive sequences as described in the original publication, we used a quantile cutoff, which does not depend on the absolute scale of $f(x)$ and performed better in our experiments. As hyper-parameters, we tuned the quantile cutoff, learning rate, batch size, discriminator and generator training epochs, the gradient penalty weight, the Gumble softmax temperature, and the number of latent variables of the generator.

## A.4   DBAS

We follow the methodology suggested by Brookes & Listgarten (2018). As hyper-parameters, we optimized the quantile for selecting training samples, learning rate, batch size, training epochs, number of hidden units of the MLP generator and discriminator, and number of latent variables. The generative model is an variational autoencoder with a multi-layer perceptron decoder. We also considered DbAs with a LSTM as generative model, which performed slightly better than a VAE on the TfBind8 problem but worse on the PdbIsing and AMP problem (see Figure 8).

## A.5   BAYESIAN OPTIMIZATION

As regressors, we considered a Gaussian process (GP) with RBF kernel on one-hot features, and an ensemble of ten fully-connected neural networks with one fully connected layer and 128 hidden units. We used the regressor output to compute the expected improvement or posterior mean acquisition function, which we maximized by gradient ascent for a certain number of acquisition steps following Killoran et al. (2017). We took the resulting $B$ unique sequences with highest acquisition function value as sequences to measure in the next round. We tuned the length scale and variance of the RBF kernel, and the learning rate, batch size, and number of training epochs of the neural network ensemble. We further tuned the number of gradient ascent steps for activation maximization.

### A.6 PPO AND DYNA PPO

We used the PPO implementation of the TF-Agents RL library (Guadarrama et al., 2018). After each round, we trained trained the agent on the collected batch of sequences for a relatively high number of steps (about 72) since it resulted in a performance increase compared with performing only a single training step. We used the adaptive KL trust region penalty, which performed slightly better than importance ratio clipping in our experiments Schulman et al. (2017). We used a policy and value network with one fully connected layer and 128 hidden units. Both networks take the current position and the $W$ last generated characters as input, which we padded at the beginning of the sequence. We set the context window $W$ to the minimum of the total sequence length and 50. As hyper-parameters, we tuned the learning rate, number of training steps, adaptive KL target, and entropy regularization. For DyNA PPO, we also tuned the maximum number of model-based optimization rounds $M$ (see Section 2.3).

### A.7 AUTOMATED MODEL SELECTION

Automatic model selection optimizes the hyper-parameters of a set of candidate models by randomized search, and evaluates each hyper-parameter configuration by five-fold cross-validation using the $R^2$ score. To account for randomness in the $R^2$ score between models due to different cross-validation splits, we used the same split for evaluating each of the models per round. We considered the following candidate models (implemented in Scikit-learn (Pedregosa et al., 2011)) and corresponding hyper-parameters:

- KNeighborsRegressor: n_neighbors
- BayesianRidge: alpha_1, alpha_2, lambda_1, lamdba_2
- RandomForestRegressor: max_depth, max_features, n_estimators
- ExtraTreesRegressor: max_depth, max_features, n_estimators
- GradientBoostingRegressor: learning_rate, max_depth, n_estimators
- GaussianProcessRegressor: with RBF, RationalQuadratic, and Matern kernel

We also considered an ensemble of 10 neural networks with two convolutional layers and one fully connected layer, and optimized the learning rate and number of training epochs.

## B DATASET DETAILS

### B.1 PROTEIN CONTACT ISING MODELS

Given a protein from the Protein Data Bank (Berman et al., 2003), we compute the energy $E(x)$ for sequence $x$ as $E(x) = \sum_i \phi_i(x_i) + \sum_{ij} C_{ij}\phi(x_i, x_j)$, where $x_i$ refers to the character in the $i$-th position of sequence $x$. $C_{ij}$ is an indicator for whether the $C\alpha$ atoms of the residues at positions $i$ and $j$ are separated by less than 6 Angstroms when the protein folds. $\phi(x_i, x_j)$ is a widely-used 'pair potential' based on co-occurence probabilities derived from the structures of real-world proteins (Miyazawa & Jernigan, 1996). The same 20 x 20 table of pair potentials is used at all positions in the sequence, and thus the difference in energy functions across proteins is dictated only by their differing contact map structure. We set the local term $\phi_i(x_i)$ to zero. In future work, it would be interesting to consider non-zero local terms.

Our experiments consider a set of qualitatively-different proteins listed at the bottom-right of Figure 1. We identify the local optima using the same procedure as in Section B.2, except without accounting for reverse complements.

### B.2 TRANSCRIPTION FACTOR BINDING SITE DATASET

We used the dataset described by Barrera et al. (2016), and min-max normalized binding affinities between zero and one. To reduce computational costs, we only considered the first replicate (REF R1) of each wild type transcription factor in the dataset, which resulted in 41 optimization targets that we used for comparing optimizers as described in Section 4.2. We extracted local optima for

each binding target as follows. First, we separated sequences into forward and reverse sequences by ordering sequences lexicographically and including each sequence in the set of forward sequences unless the set already contained its reverse complement. We then chose the 100 forward sequences with the highest binding affinity and clustered them using the hamming distance metric, where we determined the number of clusters by finding the number of PCA components required to explain 95% of variance. We then used the sequences with the highest reward per cluster and their reverse complement as local optima.

### B.3    ANITMICROBIAL PEPTIDE DATASET

We downloaded the dataset[1] provided by Witten & Witten (2019), and followed the paper for preprocessing sequences and generating non-AMP sequences as negative training samples. We additionally excluded sequences containing cysteine and sequences shorter than 15 or longer than 50 amino acids. We fit one classifier to predict if a sequence is antimicrobial towards either E.coli, S.aureus, P.aeruginosa, or B.subtilis, which we used for hyper-parameter tuning, and a second classifier for C. alibicani, which we used for hold-out evaluation. We used C. alibicani as hold-out target since its antimicrobial activity was least correlated with the activity of other pathogenes in the dataset with more than 1000 AMP sequences. We used random forest classifiers since they were more accurate (cross-validated AUC 0.99 and 0.94) than alternative models such as k-nearest neighbors, Gaussian processes, or neural networks. Since sequences are variable-length, we padded them to the maximum sequence length of 50 and extended the vocabulary by an additional end of sequence token. Tokens after the fist end of sequence token were ignored when evaluating $f(x)$.

---

[1]https://github.com/zswitten/Antimicrobial-Peptides

# C COMPARISION OF RL METHODS

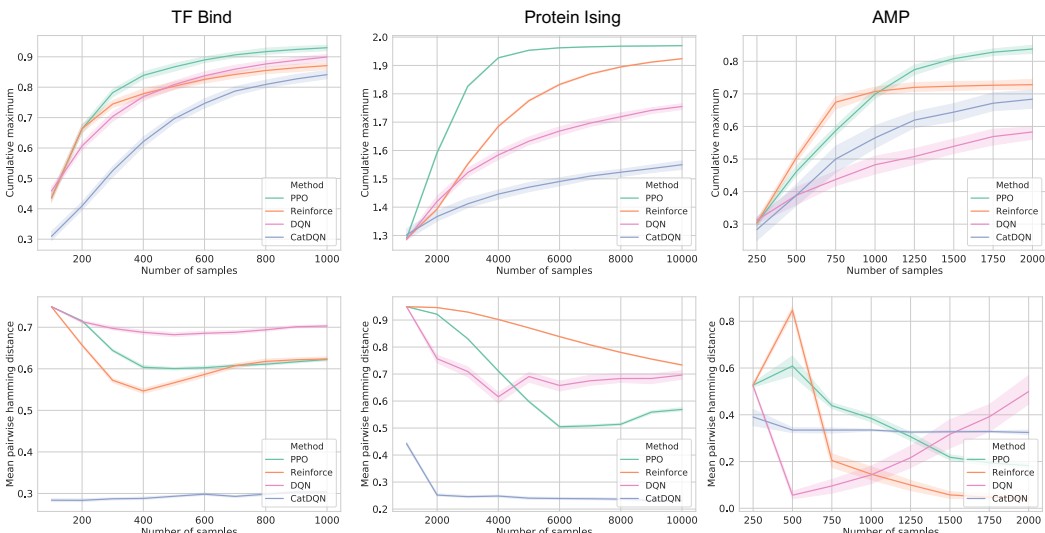

Figure 7: **Comparison of PPO against alternative RL methods.** Shown are the cumulative maximum reward and mean pairwise hamming distance for the transcription factoring binding, protein Ising, and AMP problem (Section 4).

DyNA PPO is built on PPO, which we have found to outperform other policy-based and value-based RL methods in practice on our problems. In Figure 7 we contrast the performance of PPO (Schulman et al., 2017), REINFORCE (Williams, 1992), deep Q-learning (DQN) (Mnih et al., 2015), and categorical distributional deep Q-learning (CatDQN) (Bellemare et al., 2017) on all problems considered in Section 4. We find that PPO has better exploration properties than REINFORCE, which tends to converge too soon to a local optimum. The poor performance of DQN and CatDQN can be explained by the sparse reward (the reward is only non-zero at the terminal state), such that the Bellman error and training loss for updating the Q network are zero in most states. We also found the performance of DQN and CatDQN to be sensitive to the choice of the epsilon greedy rate and Boltzmann temperature for trading-off exploration and exploitation and increasing diversity.

# D    COMPARISON OF ADDITIONAL BASELINES

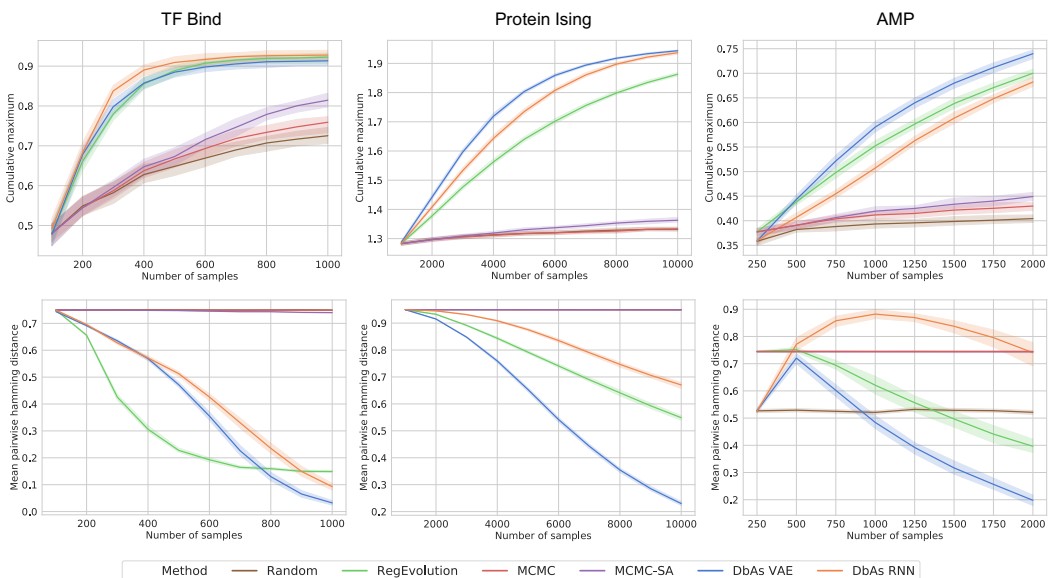

Figure 8: **Comparison of additional baselines.** We consider the performance of optimizers based on MCMC (Section A.2). Such methods are known to be effective optimizers when evaluating the black-box function is inexpensive, and thus many iterations of sampling can be performed. The focus of our experiments is on resource-constrained black-box optimization. We find that their low sample efficiency makes them undesirable for biological sequence design. We also consider DbAS with a LSTM generative model instead of a VAE with multi-layer perceptron decoder to disentangle the choice of generative model in DbAS from the overall optimization strategy. DbAs VAE outperforms DbAs RNN on all problems except for TF bind.

# E ANALYSIS OF THE EXPLORATION BONUS

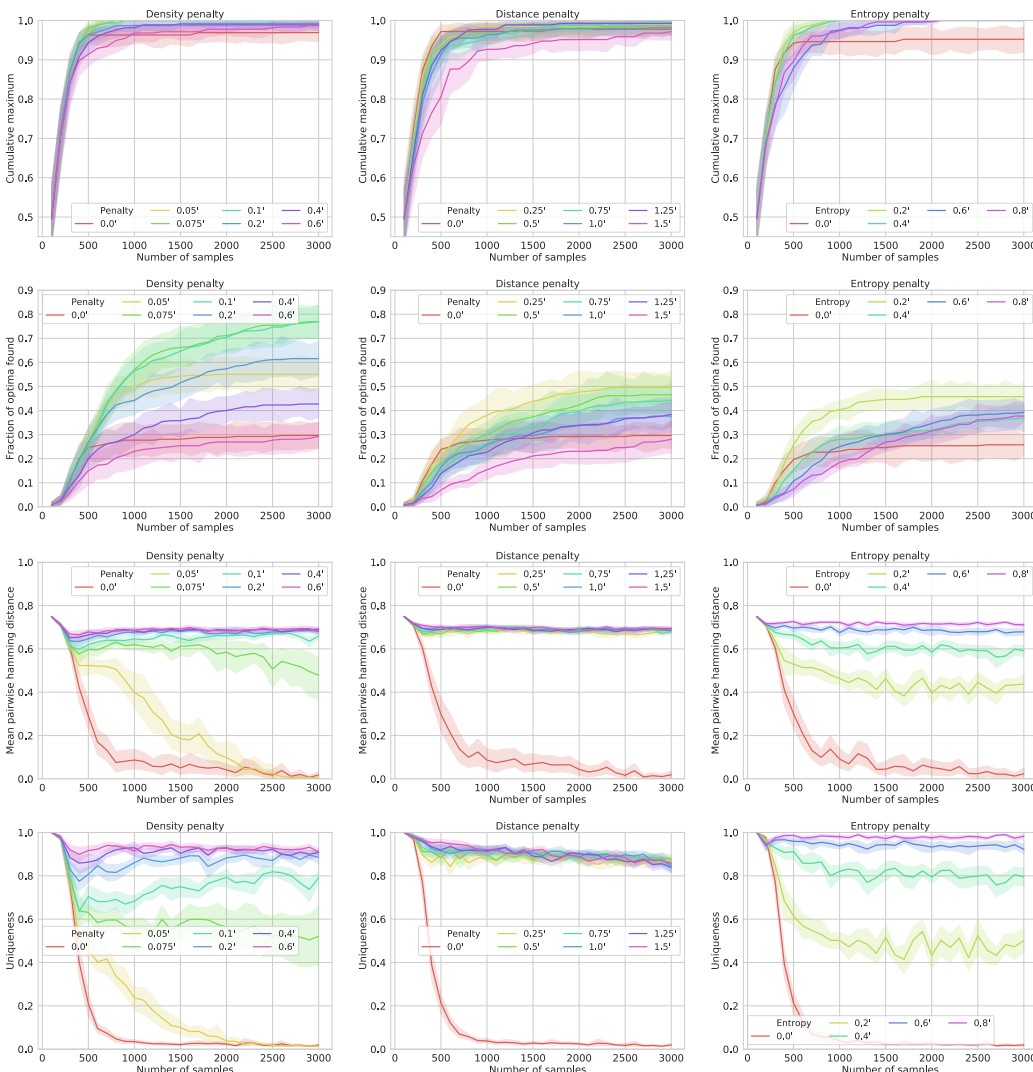

Figure 9: **Comparison of alternative approaches for promoting diversity.** Left column: The proposed density based exploration bonus as described in Section 2.4, which adds a penalty to the reward of a sequence x that is proportional to the distance-weighted number of past sequence that are less than a specified distance away from x (here edit distance one). Middle column: An alternative approach where the exploration bonus of a sequence is proportional to the distance to the nearest neighboring past sequence. Right column: standard entropy regularization. Shown are the cumulative maximum reward and alternative metrics for quantifying diversity depending on the penalty strength ($\lambda$ in Section 2.4) of each exploration approach. Without exploration bonus (penalty = 0.0; red line), PPO does not find the optimum (cumulative maximum is below 1.0) and the hamming distance and uniqueness of sequences within a batch converge to zero. PPO finds the optimal solutions and continues to generate diverse sequences by increasing the strength of any of the three exploration approaches. The density based exploration bonus is most effective in recovering all optima (second row, left plot) and enables a more fine-grained control of diversity compared to the distance based approach. Results are shown for target CRX_REF_R1 of the transcription factor binding problem.

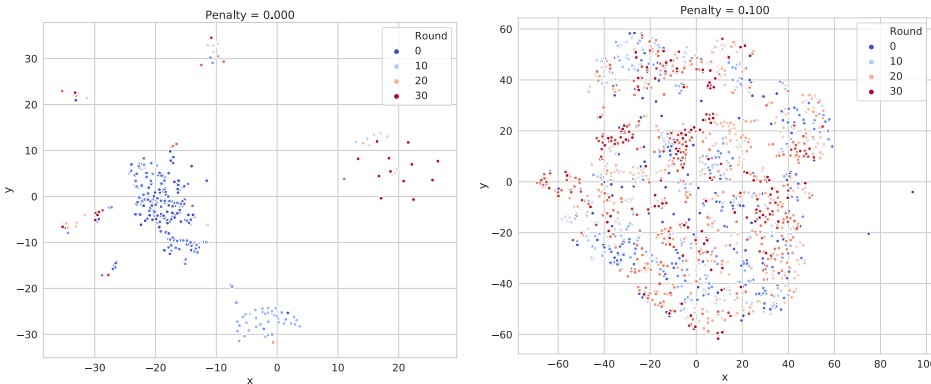

Figure 10: **tSNE embedding of proposed sequences with and without exploration bonus colored by the optimization round when they were proposed.** Without exploration bonus (penalty = 0.0), sequences cluster into few groups of low diversity at different rounds. With the exploration bonus, the diversity of proposed sequences remain high also in later rounds. Results are shown for target CRX_REF_R1 of the transcription factor binding problem.

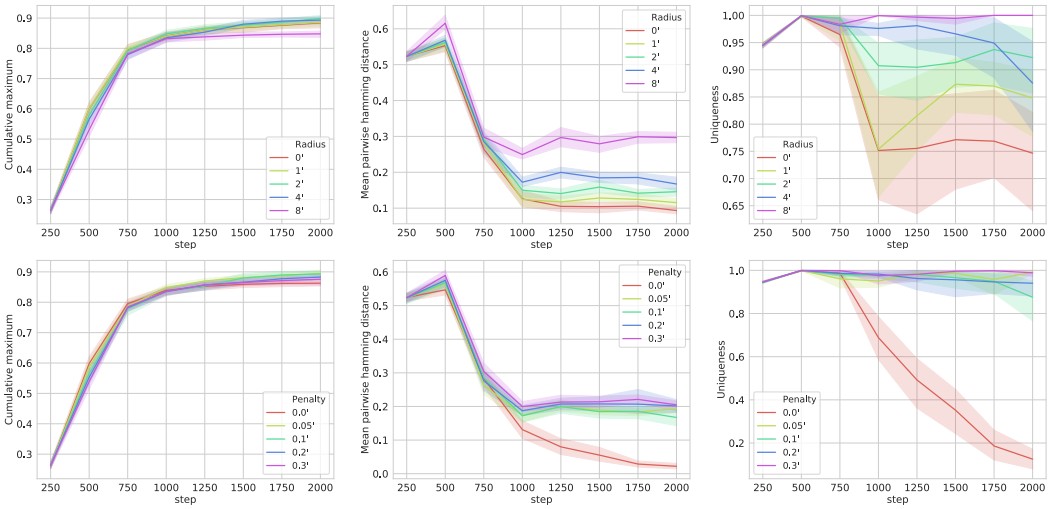

Figure 11: **Analysis of density-based exploration bonus on the AMP problem.** The top row shows the sensitivity to the distance radius $\epsilon$ and the bottom row to the regularization strength $\lambda$ (Section 2.4). Diversity correlates positively with the distance radius and regularization strength. $\epsilon = 2$ and $\lambda = 0.1$ provides the best trade-off between optimization performances (cumulative maximum reward) and diversity (mean pairwise hamming distance and uniqueness). Penalizing only exact duplicates ($\epsilon = 0$) is less effective in maintaining a high hamming distance than taking neighboring sequences into account ($\epsilon > 0$).

## F ANALYSIS OF MODEL-BASED TRAINING

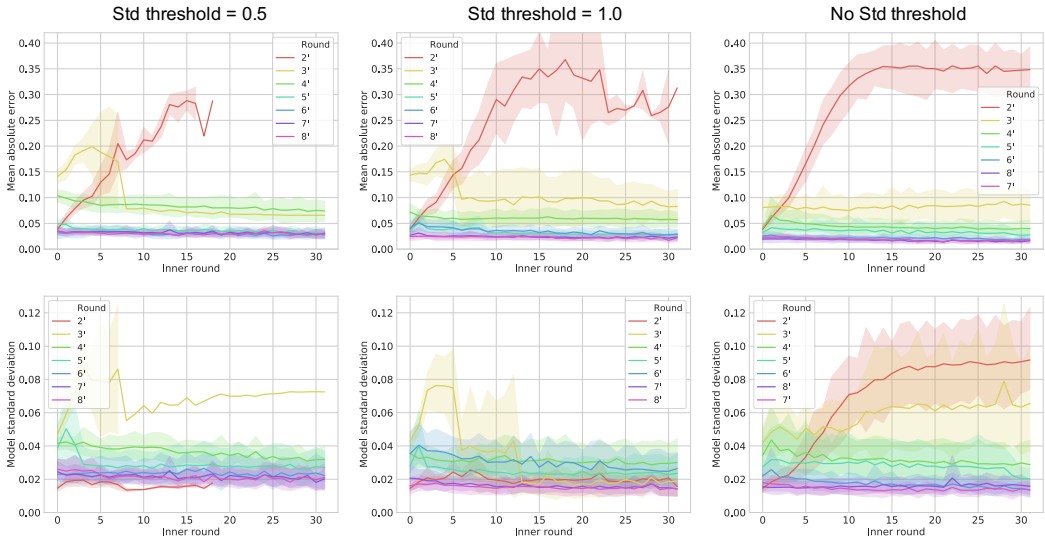

Figure 12: **Analysis of model accuracy during model based training on the AMP problem.** Shown are the mean absolute error (top row) and model uncertainty (bottom row; standard deviation) depending on the number of inner model-based optimization rounds $m$ (x-axis; see Algorithm 1) and outer optimization rounds $n$ (colors). Columns correspond to different thresholds for stopping model optimization (see Section 2.3). Without threshold (right column), the model error and model uncertainty increase rapidly after only a few inner optimization rounds. The model uncertainty is strongly correlated with the model error ($R^2 = 0.87$), and can be hence used as proxy for the unknown model error. Stopping model-based optimization as soon the the model uncertainty increases by a factor of 0.5 (left column) upper-bounds the model error and prevents DyNA PPO from performing policy updates with inaccurate model predictions.

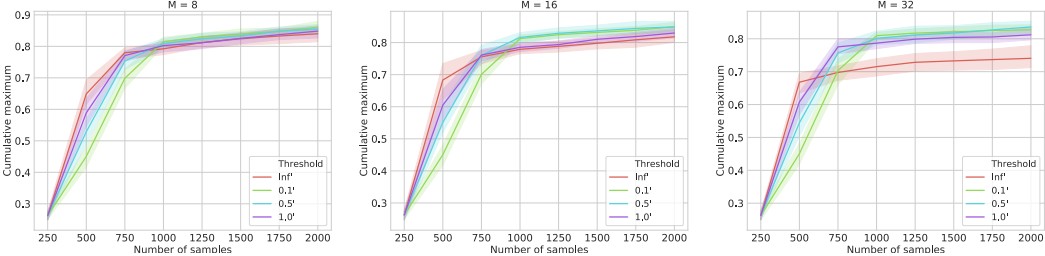

Figure 13: **Optimization performance on the AMP problem depending on the uncertainty threshold for stopping model-based optimization and the maximum number model optimization rounds M.** Without threshold (Inf; red line), DyNA PPO converges to a sub-optimal solution, in particular when the maximum number of model-based optimization rounds M is high. A threshold of 0.5 prevents a performance decrease due to inaccuracy of the model (see Figure 12).

# G    COMPARISON OF CROSS-VALIDATION SPLITTING STRATEGIES

We used the k-fold cross-validation tools in scikit-learn for performing model selection. After publication of the paper, we discovered that the default behavior in sklearn.model_selection.KFold is to not shuffle the input data but to slice them into chunks based on the input ordering.

When we switched to using random cross-validation folds, we found that the predictive accuracy of models was considerably higher than when using folds based on the data order. This led to different models being selected, which led to a slight decrease in black-box optimization performance compared to when not shuffling the input data (Figure 15).

Our data was sorted in the order in which it appeared in the optimization rounds. Hence, k-fold cross-validation without shuffling corresponds to splitting the data approximately by rounds, which favorably selects models that generalize across rounds. This is desired since samples in the same

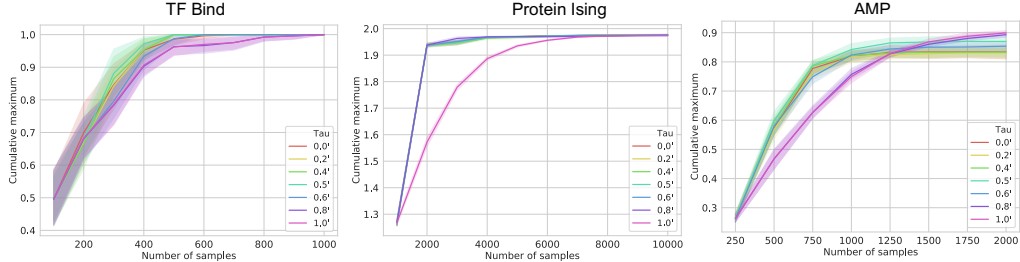

Figure 14: **Sensitivity of DyNA PPO depending on the choice of the minimum cross-validation score $\tau$ for model-based optimization.** Shown are the results for the transcription factor binding-, protein contact Ising-, and AMP problem. DyNA PPO reduces to PPO if $\tau$ is above the maximum cross-validation score of models that are considered during model selection, e.g. if $\tau = 1.0$. If $\tau$ is too low, also inaccurate models are selected, which reduces the overall accuracy of the ensemble model and optimization performance. A cross-validation score between 0.4 and 0.5 is best for all problems.

round tend to be correlated. It splits the data only approximately by rounds if the number of folds k is not equal to the number of rounds performed so far.

In response, we ran experiments using a true round-based split, which performed similarly to splitting the data approximately by rounds using sklearn.model_selection.KFold with shuffle=False. Based on the similar performance of these two slitting strategies, we did not change the experiments in the paper.

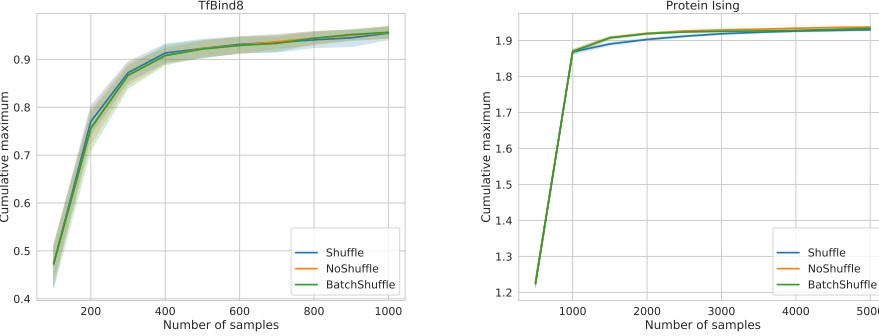

Figure 15: **Comparison of cross-validation splitting strategies.** Shown are the mean optimization trajectories over 12 TfBind8 targets and 10 Protein Ising models when splitting the input data into 5 folds with shuffling the input data (Shuffle), without shuffling the input data (NoShuffle), and splitting the input data by optimization rounds (BatchShuffle). Splitting the input data approximately by rounds (NoShuffle) or exactly by rounds (BatchShuffle) results in a performance increase on Protein Ising problems compared with splitting the data randomly (Shuffle). NoShuffle performs as well as BatchShuffle on TfBind8, and better than BatchShuffle and Shuffle on all PdbIsing targets.

