# OpenReview forum: "Model-based reinforcement learning for biological sequence design"
_ICLR.cc/2020/Conference — Accept (Poster)_

### Official Review · AnonReviewer2 · 2019-10-23
**Official Blind Review #2**

**Rating:** 6

**Review:**

Contribution
This paper apply a model-based RL algorithm, DyNA-PPO for designing biological sequences. By being model-based, this algorithm is sample efficiency compared to model-free RL algorithms. This advantage is attractive and important in the context of biological sequence design since the designed is constrained to be done in the large batch / low round settings. To further improves model efficiency, the authors reduce learning bias by quantifying the reliability and automatically selecting models of appropriate complexity via cross validation. To encourage diversity in the target distribution they also penalize the reward using a visitation-based strategy.


Clarity
Overall, the paper is well written, well motivated and well structured. The technical content is also very clear and good.


Novelty
The novelty in this work seems to be more on the applicative side (RL to optimizing DNA and protein sequences) than the method itself. I agree with the authors that most existing optimization methods are ill equipped for the large batch / low round settings and as sample efficiency becomes critically important as the number of round gets lower and their method is a good solution in such settings.

The technical novelty seems incremental as cross-validating and using a set of models under particular performance constraints does not constitutes a novel contribution. Penalizing the reward if the same sequence is seen multiple times seems decent and works well compared to the entropy regularization but it is still questionable if it is the best solution for biological sequences. Showing results when the reward is penalized using the hamming loss or the biological similarity  with previous sequences could go a long way to convince of your choice.


Experiments:
The experiments are overall well presented and seems robust given the number of replicates that was made each time.
Analyzing the model performances across different metrics: diversity, fraction of optimals, cumulative maximum helped to  understand the method and its advantages.

However, I would like to see other RL algorithms that were shown in the appendix for all those comparisons.
Including MCMC methods in the experiments will also allow to see how RL methods compared to  the sota in bioinformatics.
For the Icing dataset, you mention that it is a contribution but do not provide enough details regarding it to allow further research with it.

In a real life biological setting, the data obtained at each batch will be more likely different both in term of sequences but also in term of labels. How all your method (and the others) perform with changing distribution of data from one batch to another. Ex (e.g. 0 < y batch 1 < 100, 100 <= y batch 2 < 500, etc) ?

You tried two values for R^2 in one experiment (one positive and one negative). What happens for any other positive values (e.g 0.1. 0.2, 0.3, etc) ?


Points of improvement
Given the applicative nature of the paper and the proposed method there are few small experiments that could have been done to strengthen the manuscript (see questions and comments above).

Preliminary rating:
* weak accept *

**Experience Assessment:**

I have read many papers in this area.

**Review Assessment: Checking Correctness Of Derivations And Theory:**

N/A

**Review Assessment: Checking Correctness Of Experiments:**

I carefully checked the experiments.

**Review Assessment: Thoroughness In Paper Reading:**

I read the paper thoroughly.

---

> ### Author Response · Authors · 2019-11-15
> **Included additional baseline and ablation studies**
>
> > Penalizing the reward if the same sequence is seen multiple times seems decent and works well compared to the entropy regularization but it is still questionable if it is the best solution for biological sequences. Showing results when the reward is penalized using the hamming loss or the biological similarity  with previous sequences could go a long way to convince of your choice.
>
> Thanks for your feedback. To motivate the choice of penalizing duplicate sequences, we included an additional analysis that compares penalizing sequences based on their frequency to penalizing sequences based on the distance to the nearest neighbor and entropy regularization (see figure 9). The results show that scaling the reward by the distance also increases the hamming distance and uniqueness of sequences. However, it is less effective in finding all optima and does not provide a fine-grained control of diversity.
>
> We further generalized our described exploration bonus to not only penalize exact matches, but take all past sequences within a specified distance radius into account. We show in figure 11 that including sequences within a radius greater than zero improves exploration over only penalizing exact duplicates (radius 0). This is because exact duplicates are unlikely in case of high-dimensional problems such as the AMP problem.
>
> =============================================
> > However, I would like to see other RL algorithms that were shown in the appendix for all those comparisons.
>
> Figure 7 now compares PPO with REINFOCE, DQN, and categorical DQN on all three optimization problems and shows that PPO performs best. PPO has better exploration properties than REINFORCE, which tends to converge too soon to a local optimum. The poor performance of DQN and CatDQN can be explained by the sparse reward (the reward is only non-zero at the terminal state), such that the Bellman error and training loss for updating the Q network are zero in most states. We also found the performance of DQN and CatDQN to be sensitive to the choice of the epsilon greedy rate and Boltzmann temperature for trading-off exploration and exploitation and increasing diversity.
>
> =============================================
> > Including MCMC methods in the experiments will also allow to see how RL methods compared to  the sota in bioinformatics..
>
> Thanks for the suggestion. We expect that MCMC methods are sota on optimization problems for which we know the functional form of the objective function, but not where we are performing black-box optimization. Nevertheless, we have added MCMC and simulated annealing to our analysis. We show in figure 8 that it performs considerably worse than regularized evolution on all problems. The details of our MCMC and simulated annealing approaches are provided in appendix A.2.
>
> =============================================
> > For the Ising dataset, you mention that it is a contribution but do not provide enough details regarding it to allow further research with it.
>
> We have updated Sec B.1 to explain the details of our new 20-state Ising model and to provide additional implementation details. Please let us know if you have more questions. It would be interesting in future work to extend these Ising models to contain, for example, higher order terms.
>
> =============================================
> > In a real life biological setting, the data obtained at each batch will be more likely different both in term of sequences but also in term of labels. How all your method (and the others) perform with changing distribution of data from one batch to another. Ex (e.g. 0 < y batch 1 < 100, 100 <= y batch 2 < 500, etc) ?
>
> We agree with you that this is a challenging ML problem because the distribution over rewards is changing batch to batch. As we optimize, the rewards generally get higher. This is exactly why we adaptively perform model selection (both model type and hyper-parameters) at each round in order to make sure that we adjust to properties of each batch. Otherwise, it is difficult to find a regressor that is suitable across rounds of experiments.
>
> =============================================
> > You tried two values for R^2 in one experiment (one positive and one negative). What happens for any other positive values (e.g 0.1. 0.2, 0.3, etc) ?
>
> We have added figure 14, which shows the sensitivity of DyNA PPO to the choice of \tau on all three design problems. Choosing \tau between 0.4 and 0.5 is best on all problems. We explain the effect of choosing \tau too small or large in the figure caption. See also our response to reviewer 4.

---

### Official Review · AnonReviewer1 · 2019-10-24
**Official Blind Review #1**

**Rating:** 3

**Review:**

Designing new discrete sequences satisfying desirable properties is an important problem in molecular biology. This is a difficult combinatorial optimization problem because of the difficulty in optimizing over a combinatorially large space. The authors propose a RL based framework for this black box optimization problem.

The paper is well written, but I have questions about the efficacy of the method, particularly because I think some of these results are against weak baselines. For example, the authors don't compare against many better performing protein design methods (See: Ingraham et. al, GENERATIVE   MODELS   FOR   GRAPH-BASED   PROTEIN DESIGN, Sabban et. al, RamaNet: Computational De Novo Protein Design using a Long Short-Term Memory Generative Adversarial Neural Network). VAE based methods have worked well for designing sequences like SMILES strings, but the authors dismiss them claiming that they are better modelled as molecular graphs. While it could be true that molecules are better modeled as molecular graphs, it is not clear why methods that have worked well in a sequence based modeling using SMILES strings will not work for Protein Design. For AMP Design, again they compare with a weak baseline and don't compare with VAE based methods (like for example: Das et al. PepCVAE: Semi-Supervised Targeted Design of Antimicrobial Peptide Sequences)

**Experience Assessment:**

I have published one or two papers in this area.

**Review Assessment: Checking Correctness Of Derivations And Theory:**

N/A

**Review Assessment: Checking Correctness Of Experiments:**

I carefully checked the experiments.

**Review Assessment: Thoroughness In Paper Reading:**

I read the paper thoroughly.

---

> ### Author Response · Authors · 2019-11-15
> **Included additional baselines and discussed related work**
>
> Thank you for suggesting the important related work (Ingraham et al, Sabban et al, and VAE based generative models). We have added a discussion of them in the related work section. Below are some additional details explaining the relationship between these works and our paper.
>
> A VAE as used in PepCVAE or for generating SMILE string is not a sequence design method, but a specific generative model that can be used within a variety of design approaches. The DbAs method that appears in all three sections of our experiments employs a VAE. We have updated the text to clarify this. We have added an experiment analyzing the performance of DbAS with a VAE vs. LSTM generative model in figure 8. The LSTM performs worse than DbAs VAE on high-dimensional problems and only slightly better on TF Bind.
>
> Ingraham et addresses the problem of inverse protein folding, i.e. generating a protein sequence that folds into a given protein structure. Instead our paper addresses multi-round optimization of any blackbox functions, which requires methods that are sample efficient and generate diverse sequences--two challenges that are not addressed in Ingraham et al. Ingraham et al considers graph-conditional sequence generation and one important contribution of their paper is the architecture for encoding protein graph structures. We consider consider unconditional optimization and do not address the problem of encoding protein graph structures. In many biological sequence design problems, the structure of interest is unknown, so a generative model that conditions on structure would not be applicable.
>
> Sabban et al describes a method for generating protein graph structures, whereas our paper is about generating sequences. Generative models for protein structures is a distinct research challenge, which we do not address. It is not clear how such a model could be directly used in multi-round biological sequence design.

---

### Official Review · AnonReviewer4 · 2019-10-30
**Official Blind Review #4**

**Rating:** 6

**Review:**

In this work the authors propose a framework for combinatorial optimisation problems in the conditions that the measurements are expensive. The basic idea is to make an approximation of the reward function and then train the policy using the simulated environment based on the approximated reward function. The applications are shown in a set of biological tasks, which shows that the model performs well compared to the baselines.

The idea of learning the models of environment (or reward) and simulating the model to train the policy is not novel (e.g., https://arxiv.org/pdf/1903.00374.pdf). Similarly, in terms of formulating the discrete search problem as a reinforcement-learning problem, again there are similar works in the past, which are cited in the paper, but the combination of these two is novel to my knowledge; having said this the paper should discuss relevant works such as the one above.

The experiments seem convincing to me overall, however, I have the following concerns:

- The performance of the model seems similar to PPO in the large state-spaces (section 4.3), which somehow is disappointing.

- The performance of the model seems very sensitive to the choice of \tau (Figure 6 right), which is set to 0.5, but it is not mentioned how this parameter is chosen (or at least I couldn’t find it) and how much the performance of the model in the other experiments is affected by the choice of this parameter.



**Experience Assessment:**

I have read many papers in this area.

**Review Assessment: Checking Correctness Of Derivations And Theory:**

I assessed the sensibility of the derivations and theory.

**Review Assessment: Checking Correctness Of Experiments:**

I assessed the sensibility of the experiments.

**Review Assessment: Thoroughness In Paper Reading:**

I read the paper at least twice and used my best judgement in assessing the paper.

---

> ### Author Response · Authors · 2019-11-15
> **Addressed all concerns**
>
> =============================================
> > The performance of the model seems similar to PPO in the large state-spaces (section 4.3), which somehow is disappointing.
>
> We agree, and this motivated us to do additional research on this problem and to make key updates to the paper.  We found that the model error on this problem increased rapidly during model-based training in the first few rounds (figure 12) and that the policy was hence trained with incorrect rewards, which decreases the overall optimization performance (figure 13). We also found that the model uncertainty (quantified by the standard deviation of the ensemble predictions) is strongly correlated with the model error (figure 12), and can therefore be used as a proxy for the reliability of the model. We do not know the model error at policy optimization time, but we do know its uncertainty.  We therefore extended DyNA PPO to not train the policy with the model for a fixed number of rounds, but to stop model-based training as soon as the model standard deviation increased by a certain factor, i.e. the model starts to become unreliable. This prevents training the policy with incorrect rewards and improves optimization performance.
>
> We could further improve the performance by generalizing our reward function to not only penalize exact duplicates, but by the number of previously proposed sequences that are within a certain distance radius around the proposed sequences. We describe the extended density-based exploration bonus in section 2.4 and compare it with alternative approaches in  figures 9-11.
>
> As a result, DyNA PPO performs now also best on the AMP problem (figure 6, left).
>
>
> =============================================
> > The performance of the model seems very sensitive to the choice of \tau (Figure 6 right), which is set to 0.5, but it is not mentioned how this parameter is chosen (or at least I couldn’t find it) and how much the performance of the model in the other experiments is affected by the choice of this parameter.
>
> We agree robustness to the choice of \tau is important. We have updated section 2.4 to clarify that we treat \tau as a tunable hyper-parameter. We have also added figure 14, which shows that performance is relatively insensitive to the choice of \tau, in particular if most models that are considered during model selection are accurate. In case of the protein contact map Ising problem, for example, the cross-validation score of most models is above 0.7, i.e. choosing any threshold below 0.7 does not decrease model performance. The performance is more sensitive to the choice of \tau if some models that are considered during model selection are inaccurate, for example in the case of the AMP problem. In this case, inaccurate models will be selected when choosing a small \tau, and the resulting ensemble model will be hence less accurate. DyNA PPO reduces to PPO when choosing \tau close to 1.0 since models have almost never a cross-validation score of 1.0 and are hence not used for model-based optimization.
>
> =============================================
> > The paper should discuss relevant works such as the one above.
>
> Thanks for the pointers. We added additional citations and extended our discussion of existing model-based approaches in the related work section.

---

### Author Response · Authors · 2019-11-15
**Response to the reviewers**

We would like to thank all reviewers for evaluating our manuscript. We have tried to address all concerns in a proper way and believe that our paper has improved considerably. In summary, we made the following changes:

1. We generalized our proposed diversity promoting reward function to take all neighbors within a specified distance radius into account instead of only exact duplicates, which we show improves exploration on higher dimensional problems such as the AMP problem. See figures 9-11 and section 2.4.

2. We extended DyNA PPO to stop model-based optimization as soon as the model uncertainty increases by a certain factor instead of training the policy for a fixed number of rounds. This is motivated by our observation that the model uncertainty is strongly correlated with the model error. See figures 12-13 and section 2.3.

3. DyNA PPO performs now best also on the AMP problem due to point 1 and 2 (see figure 6).

4. We changed the vocabulary size of the proposed protein contact Ising model from two to twenty, the number of amino acids, to make it a more realistic protein design problem. We also discuss the future research and why this problem is of value for the protein design community. See section 4.1 and B.1.

5. We included MCMC, simulated annealing, and DbAs with a LSTM instead of VAE as generative model. See section A.2 and figure 8.

We responded in detail to all comments. We would be happy to make further corrections if necessary. Overall, we believe that our paper is interesting for both applied computational biologists and machine learning researchers. We hope that our proposed optimization problems and methods will inspire future research on blackbox optimization, generative modeling, and uncertainty quantification, with the ultimate goal to improve drug design or make manufacturing processing more sustainable.

---

### Decision · Program_Chairs · 2019-12-19

**Decision:**

Accept (Poster)

**Comment:**

The paper proposes a model based proximal policy optimization reinforcement learning algorithm for designing biological sequences. The policy of for a new round is trained on data generated by a simulator. The paper presents empirical results on designing sequences for transcription factor binding sites, antimicrobial proteins, and Ising model protein structures.

Two of the reviewers are happy to accept the paper, and the third reviewer was not confident. The paper has improved significantly during the discussion period, and the authors have updated the approach as well as improved the presented results in response to comments raised by the reviewers. This is a good example of how an open review process with a long discussion period can improve the quality of accepted papers.

A new method, several nice applications, based on a combination of two ideas (simulating a model to train a policy RL method, and discrete space search as RL). This is a good addition to the ICLR literature.